# TUT-DIS3L2 is a mammalian surveillance pathway for aberrant structured non-coding RNAs

Dmytro Ustianenko[1,†], Josef Pasulka[1,†], Zuzana Feketova[1,†], Lukas Bednarik[1], Dagmar Zigackova[1,2], Andrea Fortova[1], Mihaela Zavolan[3] & Stepanka Vanacova[1,*]

## Abstract

Uridylation of various cellular RNA species at the 3′ end has been generally linked to RNA degradation. In mammals, uridylated pre-let-7 miRNAs and mRNAs are targeted by the 3′ to 5′ exoribonuclease DIS3L2. Mutations in DIS3L2 have been associated with Perlman syndrome and with Wilms tumor susceptibility. Using *in vivo* cross-linking and immunoprecipitation (CLIP) method, we discovered the DIS3L2-dependent cytoplasmic uridylome of human cells. We found a broad spectrum of uridylated RNAs including rRNAs, snRNAs, snoRNAs, tRNAs, vault, 7SL, Y RNAs, mRNAs, lncRNAs, and transcripts from pseudogenes. The unifying features of most of these identified RNAs are aberrant processing and the presence of stable secondary structures. Most importantly, we demonstrate that uridylation mediates DIS3L2 degradation of short RNA polymerase II-derived RNAs. Our findings establish the role of DIS3L2 and oligouridylation as the cytoplasmic quality control for highly structured ncRNAs.

**Keywords** DIS3L2; ncRNAs; RNA surveillance; TSSa; uridylation
**Subject Categories** RNA Biology
**The EMBO Journal (2016) 35: 2179–2191**

## Introduction

Post-transcriptional modification of RNAs, by oligonucleotide addition at the 3′ termini (tailing) or by base modification, has a key function in embryogenesis, cell differentiation, circadian rhythm, immunity, and disease (reviewed in Lee *et al*, 2014). Uridine (U)-tailing generally results in RNA degradation (Lee *et al*, 2014). In mammalian cells, the two best-characterized roles of U-tailing are in the regulation of let-7 miRNA processing (Heo *et al*, 2008, 2009, 2012; Hagan *et al*, 2009; Nam *et al*, 2011; Thornton *et al*, 2012) and the degradation of replication-dependent histone mRNAs in the late S-phase of the cell cycle (Mullen & Marzluff, 2008; Schmidt *et al*, 2011; Su *et al*, 2013; Slevin *et al*, 2014). Comprehensive profiling of

RNA modifications requires targeted approaches, which only recently have been developed. Initial transcriptome-wide studies revealed the presence of one to three uridine residues on several non-coding RNAs in human ES cells (Choi *et al*, 2012), and more recently, monouridylation of shortened poly(A) tails was implicated in mRNA turnover in yeast and human cells (Rissland & Norbury, 2009; Malecki *et al*, 2013; Lim *et al*, 2014). However, non-templated uridylation does not always lead to RNA degradation. On the contrary, monouridylation of certain pre-miRNAs facilitates DICER processing (Heo *et al*, 2012) and oligouridylation of U6 snRNA is an integral part of U6 maturation (Lund & Dahlberg, 1992; Tazi *et al*, 1993).

At least four mammalian enzymes known as terminal uridyl-transferases (TUTs), TUT1/4/7 and GLD-2, have the potential to add non-templated UMPs to 3′ RNA termini (reviewed in Lee *et al*, 2014; Scott & Norbury, 2013). For this activity, they often associate with co-factors. For example, TUT4 and TUT7 processivity is enhanced upon association with LIN28A/B (Hagan *et al*, 2009; Heo *et al*, 2009; Thornton *et al*, 2012), whereas GLD-2 forms a complex with GLD-3 (Kwak *et al*, 2004). In contrast to the family of TUTases, which seems to be relatively large, only two nucleases that specifically target uridylated tails have been found to date in mammalian cells. They are ERI1/HEXO1, which acts on uridylated histone mRNAs to induce replication-dependent decay (Hoefig *et al*, 2013) and DIS3L2, which degrades uridylated precursors of let-7 miRNA (Chang *et al*, 2013; Ustianenko *et al*, 2013). Recent studies suggested a broader spectrum of DIS3L2 targets in the context of various stresses. Upon viral infection, TUTase and DIS3L2 are involved in template-dependent miRNA degradation (TDMD) (Haas *et al*, 2016) and in the decay of improperly processed ncRNAs (Eckwahl *et al*, 2015). Moreover, DIS3L2 targets uridylated mRNA cleavage products in apoptotic cells (Abernathy *et al*, 2015; Thomas *et al*, 2015). The crystal structures of mouse and yeast DIS3L2 have revealed unique residues that allow for the specific recognition of the uridine stretches as well as for its ability to process structured substrates, which is due to the specific shape of its RNA binding funnel (Faehnle *et al*, 2014). Interestingly, DIS3L2 is highly processive on structured RNA substrates *in vitro* (Lubas *et al*, 2013; Ustianenko *et al*, 2013).

1 CEITEC-Central European Institute of Technology, Masaryk University, Brno, Czech Republic
2 National Centre for Biomolecular Research, Faculty of Science, Masaryk University, Brno, Czech Republic
3 Biozentrum, University of Basel and Swiss Institute of Bioinformatics, Basel, Switzerland
*Corresponding author. Tel: +420 549495042; E-mail: stepanka.vanacova@ceitec.muni.cz
†These authors contributed equally to this work

Several members of these RNA surveillance pathways have been linked to human disease. Elevated expression of LIN28A and TUT4 has been observed in several types of carcinomas (Hallett & Hassell, 2011; Zhou *et al*, 2013). Loss-of-function germline mutations in the *DIS3L2* gene lead to the Perlman syndrome (Astuti *et al*, 2012). Disruption of *DIS3L2* and mutations in the microprocessor complex components in these patients also seems to increase the incidence of Wilms and bilateral tumors (Morris *et al*, 2013; Wegert *et al*, 2015). In cellular models, impairment of DIS3L2 function leads to severe cell cycle defects (Astuti *et al*, 2012). The mechanisms causing these DIS3L2 mutation-linked phenotypes are currently unknown.

In a previous study (Ustianenko *et al*, 2013), we have demonstrated that the catalytic site mutant of DIS3L2, D391N, efficiently binds uridylated RNAs *in vitro* and *in vivo*. In this work, we used D391N as bait to comprehensively identify the *in vivo* targets of DIS3L2 by cross-linking and immunoprecipitation followed by sequencing (CLIP-seq) in HEK293T-Rex cells. We thereby identified an extensive set of uridylated RNAs including non-coding RNAs such as the small nuclear (sn)RNAs, ribosomal (r)RNAs, transfer (t)RNAs, long non-coding (lnc)RNA, vault RNAs, Y RNAs, micro (mi)RNAs, mRNAs, and transcription start site-derived RNAs from protein-coding gene loci. Together with the biochemical evidence presented here as well as what we previously demonstrated, these data indicate that TUT-DIS3L2 are part of a general mechanism of cytoplasmic RNA surveillance and degradation in mammalian cells.

## Results

### CLIP-seq identification of uridylated RNAs bound by mutant DIS3L2

To identify DIS3L2 RNA targets, we performed CLIP-seq analysis with the catalytically inactive DIS3L2 mutant, D391N (Ustianenko *et al*, 2013), stably integrated in the HEK293T-Rex cell line (Fig 1A). Initial inspection of the sequenced reads revealed that more than 10% possessed 3′-terminal oligo(U) stretches of median length 6–8 nucleotides (Fig 1B). Whereas homomeric stretches of 1–3 A, C, or G nucleotides occurred with similar frequencies at the end of the reads, longer homomeric stretches consisted only of uridines (Fig 1B). Consistent with the post-transcriptional modification of DIS3L2 targets, 70% of the reads did not map to the human genome (hg19). Thus, to comprehensively annotate these targets, we

trimmed the oligo(U) tails from the reads and remapped them to the genome (Fig EV1A). Functional annotation of the reads that had clear evidence of non-templated uridylation revealed that the DIS3L2 D391N binds a broad spectrum of U-tailed (U+) RNAs (Figs 1C and EV1B, and Table EV1). The highest percentage (26%) of U+ reads belonged to the miscellaneous RNA (miscRNA) category, which includes vault RNA, 7SK, Y RNAs, and RNaseP. The other U+ reads mapped to Pol I rRNA (20%), snRNA (16%), mRNA (11%), 5S rRNA (10%), tRNA (6%), miRNA (7%), snoRNA (2%), and lincRNA (< 0.5%) (Figs 1C and EV1B). Although the oligo(U) tails were typically 6–8 nucleotides in length (Fig 1B), the average length differed between the target classes (Fig EV1C). Whereas for many classes of RNAs, the U+ reads mapped in the proximity of the 3′ end of the mature molecule (Fig 1C), in some snoRNAs, mRNAs and lincRNAs U+ reads mapped in the proximity of their 5′ ends (Fig 1C and see the last paragraph of the Results for more details).

The majority of DIS3L2-bound miRNAs belonged to the let-7 family (Fig 1D and Table EV1), in line with the previously reported role of TUT-DIS3L2 in pre-let-7 turnover (Hagan *et al*, 2009; Chang *et al*, 2013; Ustianenko *et al*, 2013). Interestingly, a modified mapping protocol (see Materials and Methods) revealed chimeric reads corresponding to ligated 5′ and 3′ arms of pre-let-7 in our CLIP-seq data (Fig 1E and F), which demonstrates that DIS3L2 binds to folded pre-miRNAs *in vivo*. Our data further revealed an expanded repertoire of U+ pre-miRNA targets, most of which lack an identifiable LIN28A binding motif (Table EV1).

To determine whether DIS3L2 affects the stability of its targets, we compared the transcriptome of D391N-expressing and the control HEK293T-Rex cells by RNA-seq. Although there was an overall small decrease in abundance of RNAs in DIS3L2 D391N-expression cells, the CLIPed targets were not distinguishable from non-targets in their pattern of expression change upon D391N overexpression (Fig 1G). To summarize, we demonstrated that 3′-terminal RNA uridylation is a general process relevant for the metabolism of most classes of RNAs, produced by all three nuclear RNA polymerases (RNAPs).

### The TUT-DIS3L2 surveillance (TDS) targets highly structured aberrant ncRNAs

More than 70% of the CLIP reads originated in ncRNAs other than miRNAs such as 5S rRNA, snRNAs, snoRNAs, RNaseP, vault RNA, and Y RNAs (Figs 2A and EV1B). Significantly, the U+ reads mapped often to non-functional transcripts, such as pseudogenes or

**Figure 1. DIS3L2 CLIP identifies broad repertoire of uridylated RNAs.**

A SDS–PAGE analysis of the D391N DIS3L2-RNA complexes. RNAs were radioactively labeled at the 5′ end, and DIS3L2-bound RNAs were detected by autoradiography. The negative control was performed with HEK293T-Rex cells (Control)

B DIS3L2 CLIP reads possess oligo(U) 3′ termini. Histograms of reads with 3′-terminal stretches of identical nucleotides. Only oligo(U) tails longer than four nucleotides are observed.

C The untemplated uridylation is mostly found in the proximity of mature 3′ RNA ends. The heat map shows the coverage of various RNA types by uridylated reads. RNA class and number of uridylated genes/transcripts for each category are indicated at the left. The length of each mature RNA was scaled to 100%.

D List of DIS3L2-bound uridylated miRNAs with the number of uridylated reads identified in all three replicate CLIPs. Members of the let-7 family are shown in bold.

E Identification of hybrid reads consisting of the 5′ and 3′ arm of pre-let-7 miRNA. Profile of read coverage along the pre-let-7 genomic locus is indicated above.

F Secondary structure of pre-hsa-let-7i, with both 5p and 3p mature products highlighted in bold. The intramolecular ligation position is indicated in dash line between nt 28 and 61. It corresponds to cleavage positions of RNaseT1, which leaves 3′-terminal guanosine residues.

G D391N DIS3L2 overexpression does not cause abundance changes of CLIPed uridylated RNAs. Scatter plot of transcript abundances estimated from RNA-seq from the cell line overexpressing D391N DIS3L2 and control HEK293T-Rex cells. CLIPed transcripts are marked in white.

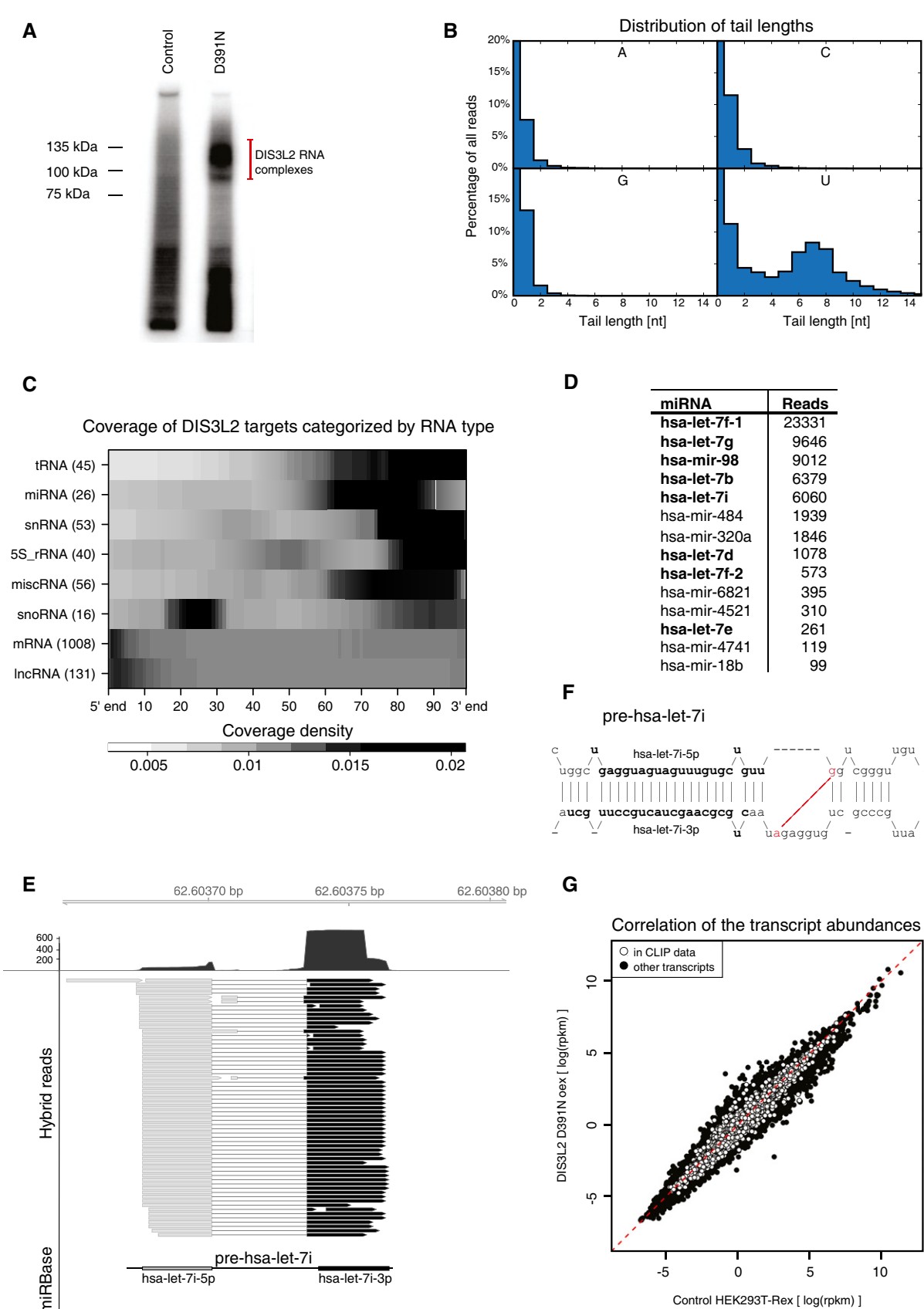

**Figure 1.**

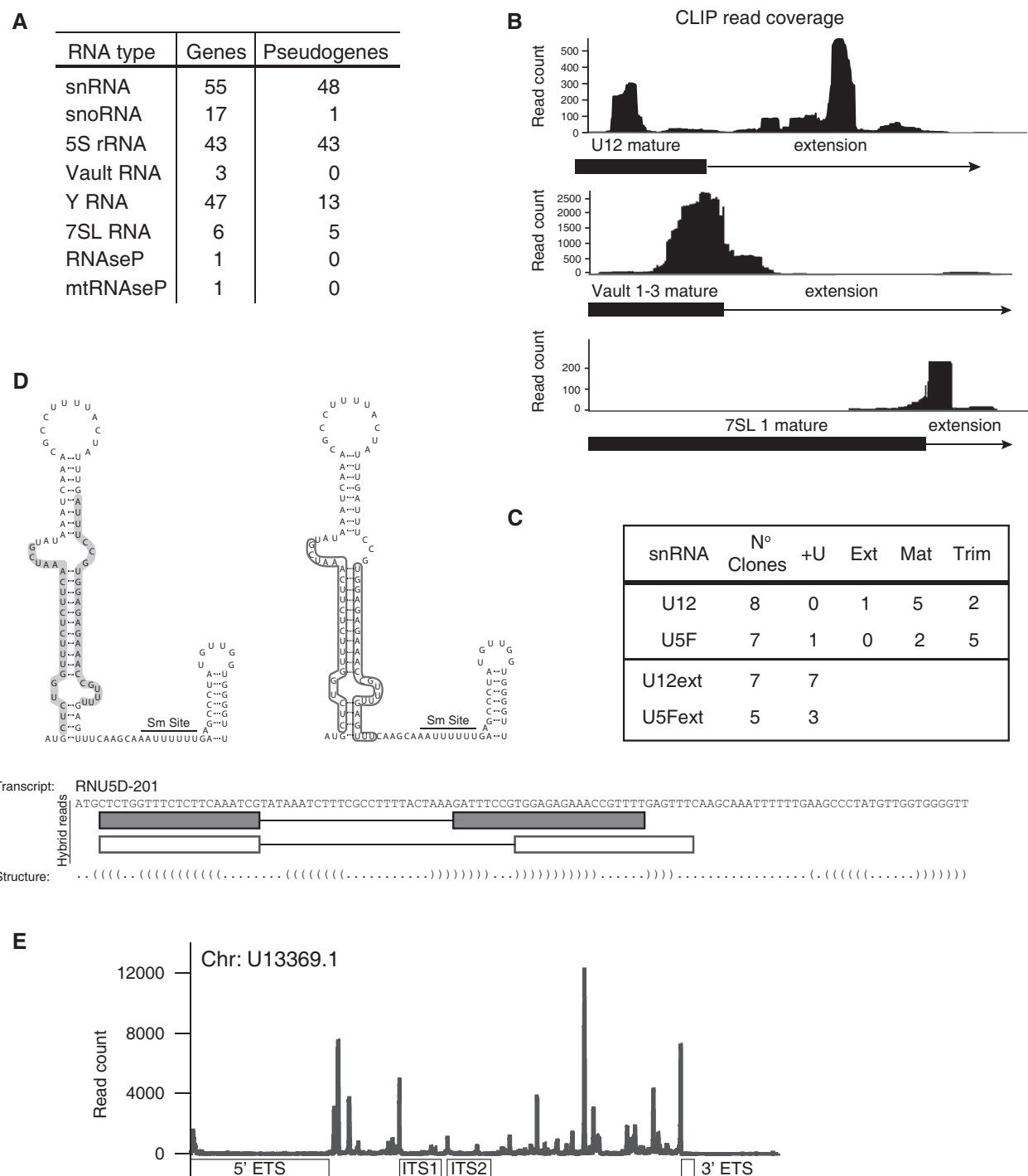

Figure 2. TUT-DIS3L2 is a surveillance pathway for structured ncRNAs produced by all three RNA polymerases.

A    Types of uridylated ncRNAs identified by DIS3L2 CLIP-seq.
B    U+ read coverage in downstream regions of mature U12 snRNA, vault1, and 7SL ncRNAs. The full dark box represents mature ncRNA, and the thin line indicates the region downstream of the mature 3′ end (extension).
C    Quantification of DIS3L2 D391N RIP analysis of U12 and U5D snRNAs showing the number of snRNA clones with mature (Mat), extended (ext), or trimmed (Trim) 3′ end. +U is the number of clones with untemplated 3′ oligo(U) ends. For more details, see Appendix Fig S2.
D    An example hybrid reads (produced by intrastrand ligation of RNaseT1 products during CLIP protocol) for RNU5D snRNA.
E    U+ reads mapped to the rDNA locus by using the artificial chromosome U13369.1.

improperly processed forms (ENSEMBL Release 78, Homo Sapiens, 2014, Fig 2A and Table EV1). In several types of ncRNAs, such as snRNAs, vault, or 7SL RNAs, the U+ reads also mapped to regions downstream of the 3′ mature termini indicating the presence of uridylated 3′-extended RNAs (Fig 2B and Appendix Table S1). We confirmed these findings on individual snRNAs by RNA immunoprecipitation (RIP) and subsequent sequencing which revealed that DIS3L2 binds trimmed as well as extended forms of snRNAs (Figs 2C and EV2A–C). The mature and partially trimmed snRNAs mostly lacked uridylation, whereas the 3′-extended forms contained U+ extensions (Fig 2C and EV2B and C). Similar to pre-let-7, we observed chimeric reads corresponding to known secondary structures on these ncRNAs (Fig 2D and Table EV1). As the presence of stable secondary structures appears to be the unifying feature of DIS3L2-bound RNAs, we propose that the TUT-DIS3L2 surveillance (TDS) is responsible for monitoring and removal of aberrant structured ncRNAs.

For RNA polymerase I-synthesized rRNAs, most of the U+ reads mapped to the mature rRNA regions, and only few aligned to 5′ and 3′ ETS as well as ITS1 and ITS2 (Fig 2E). Because DIS3L2 also associates with polysomes (Lubas *et al*, 2013), this may suggest that TDS plays a role in general rRNA turnover.

## TDS targets precursor and aberrant forms of tRNAs

Uridylated aberrant pre-tRNAs from host cells have been recently detected in retroviral particles (Eckwahl *et al*, 2015). U+ tRNA-derived fragments were highly abundant in our D391N CLIP-seq data as well (Fig EV1B), mapping to tRNA forms truncated within the T-loop or to the 3′-end tRNA trailers. These extensions are typically cleaved by tRNaseZ (Fig 3A and B). We validated these results by RIP with the WT and D391N DIS3L2. Northern blot analysis of D391N-precipitated tRNAVal revealed both slower and faster migrating species compared to mature tRNA signal (Fig 3C). Interestingly, the wild-type DIS3L2 co-precipitated shorter forms of tRNAs of about 24–30 nt in length. These could potentially represent the end-products of the DIS3L2 degradation. The DIS3 protein did not precipitate any tRNAs above the background signal, suggesting that exosome is not involved in tRNA surveillance in this context. In agreement with previous reports (Kumar *et al*, 2014), 3′-derived tRNA fragments were immunoprecipitated by AGO2 (Fig 3C). We further validated the results of the bioinformatics analysis by Sanger sequencing of cDNA clones for Leu(TAG) and Val(CAC) tRNAs prepared from the RIPed RNAs. Five out of 20 tRNALeu-derived clones mapped to the genomic locus without apparent oligo(U) tails, whereas the remaining 15 clones corresponded to 3′-truncated

tRNALeu possessing untemplated stretches of up to 12 uridines (Fig 3D). We obtained similar results for another tRNA, tRNAVal (Fig 3D). Akin to pre-let-7 and snRNAs, the sequencing data contained several hybrid reads consisting of two distinct tRNA regions (Fig 3E). Interestingly, 5′ arms of these hybrids often corresponded to 5′ tRNA trailers (Fig 3E).

To test which of the known TUTases might be responsible for the uridylation of tRNAs, we compared the *in vitro* uridylation activities of purified TUT1, TUT4, and TUT7 using the tRNA fragment (tRF) as a substrate (Fig EV3A). Whereas the activity of TUT1 was very weak, TUT7 catalyzed addition of long poly(U) tails. TUT4 modified the tRF with 10–20 UMPs, which most closely resembled the oligo(U) tails identified on RNAs in our CLIP data (Fig EV3A). Moreover, the TUT4 activity enhanced the DIS3L2 degradation of tRFs *in vitro* (Fig EV2B). These results suggested that TUT4 might be one of the enzymes acting in the TDS pathway. However, more extensive studies *in vivo* are needed to reveal the involvement of the individual TUTases in this surveillance pathway.

## Short promoter proximal RNAP II transcripts are degraded by TDS

We next examined the position of U+ mRNA reads with respect to the coding regions. We observed a striking pattern of uridylated reads mapping to either 5′ or 3′ UTRs (Fig 4A). All U+ reads mapping to 3′ UTRs originated from terminal stem-loops of histone mRNAs (Fig 4B), consistent with previous reports on the role of uridylation in histone mRNA turnover (Mullen & Marzluff, 2008; Schmidt *et al*, 2011; Su *et al*, 2013; Slevin *et al*, 2014). However, our data revealed novel uridylated 5′ mRNA fragments (U+ 5′ mRf), which corresponded to a variety of mRNAs (Table EV1). A large fraction of U+ 5′ mRfs extended from the first exon to the first intron, which indicated that uridylation occurred on unspliced pre-mRNAs (Table EV1, exon-intron sheet). We were able to validate the existence of U+ 5′ mRf independently of DIS3L2 binding by RT–PCR and sequencing of low molecular weight RNA isolated from control HEK293T-Rex cells (Figs 4C and EV4A and B). We analyzed 30 clones of the OAT 5′ mRfs PCR products from each, the control and D391N overexpressing cells. Sequencing analysis revealed that more than 60% of fragments in D391N-oex cells contained untemplated oligo(U) tails (Fig 4C), whereas only one clone from the control 293T cells had oligo(U) extension longer than three Us (Fig EV4A). This demonstrated that overexpression of inactive DIS3L2 leads to U+ 5′ mRfs upregulation. To further investigate the role of DIS3L2 in the turnover of U+ 5′ mRfs, we prepared DIS3L2 knockout (KO) cell line by using the CRISPR-Cas9 approach (Ran

**Figure 3.  TUT-DIS3L2 pathway targets precursor and aberrant forms of tRNAs.**

A   Schematic representation of the pre-tRNA secondary structure.

B   Uridylated tRNA reads overlap mainly with 3′ trailers, T-loop, and anticodon loop regions. Heat map showing the coverage of uridylated reads on pre-tRNAs. The pre-tRNA subregions (as indicated in A) are in small letters on the *x*-axis.

C   Northern blot analysis of RNAs immunoprecipitated with Flag-tagged WT (DIS3L2) and D391N (MUT) DIS3L2, AGO2, and DIS3, respectively. The blots were probed with radioactively labeled probes for 5′ and 3′ ends of mature tRNAVal, respectively.

D   Sequencing of MUT DIS3L2-bound tRNAVal and tRNALeu confirmed the uridylation status of the tRNA fragments identified by CLIP-seq. Genomic sequence for each tRNA is indicated at the top with the 3′ trailer region separated by a vertical line. Non-templated uridine residues are in red.

E   Identification of hybrid reads (produced by intrastrand ligation of RNaseT1 products during CLIP protocol) mapping to tRNAPhe(GAA) and tRNAAsn(GTT) genes. To the right, the schematic representation of selected hybrid reads with corresponding color-code.

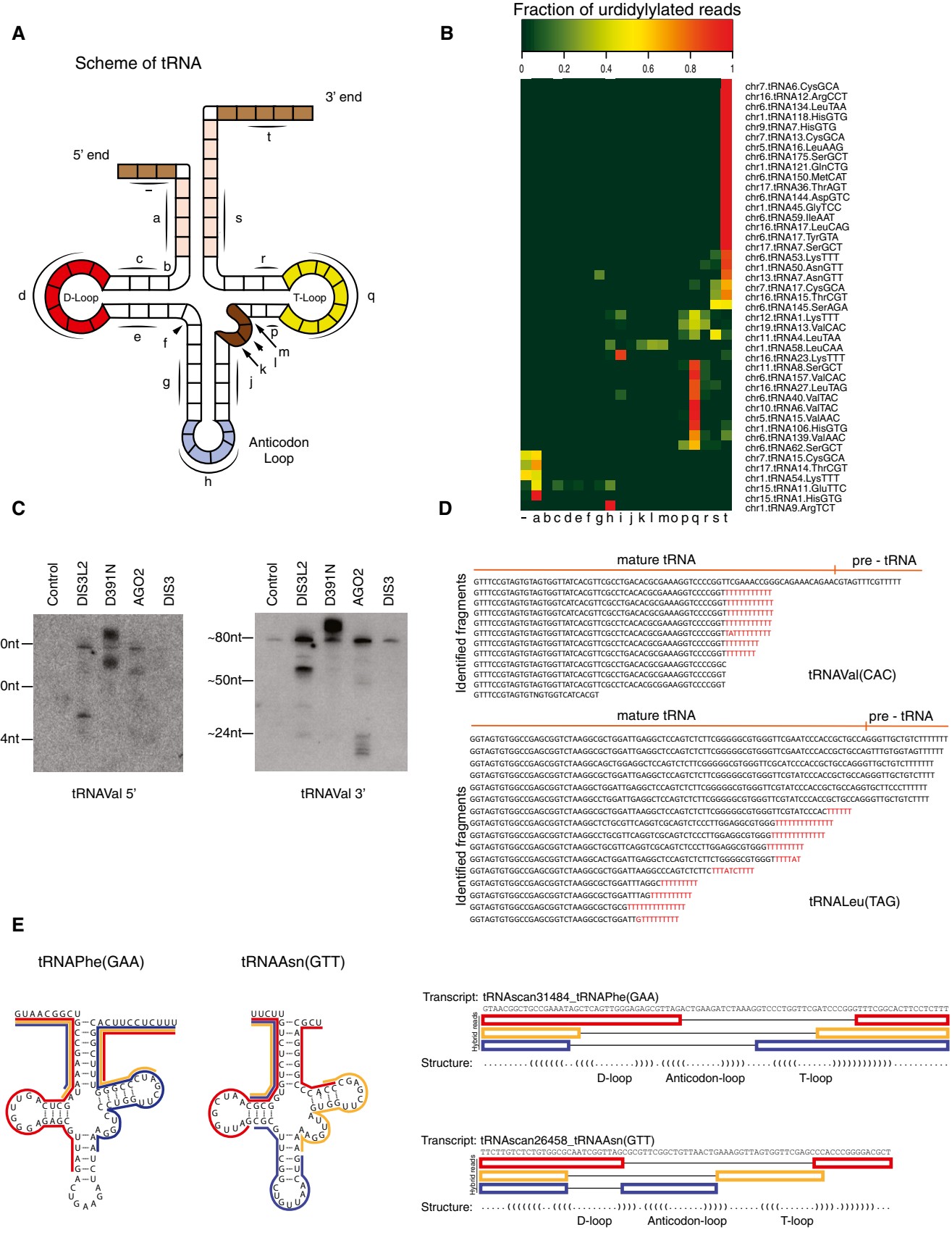

**Figure 3.**

*et al*, 2013; Hsu *et al*, 2014). To test to what extent is the enzymatic activity involved in the turnover of U+ RNAs, we stably reintroduced the wild-type (WT) or D391N mutant DIS3L2 to the KO cell line. As loading controls, we used PCR products of amplification products of the mature regions of 5S rRNA and 7SL RNA, because correctly processed functional ncRNAs do not appear to be targets of TDS. The levels of mature 5S rRNA and 7SL RNA remained unchanged in all cell lines tested. Notably, DIS3L2 depletion by KO resulted in accumulation of OAT and RPL12 U+ 5′ mRfs as well as upregulation of the 3′-extended forms of U12 snRNA (RNU12ext) (Fig 4D). This upregulation was strongly augmented by the additional overexpression of mutant DIS3L2 in KO cells. Conversely, reintroduction of the WT DIS3L2 resulted in reduction of the aberrant U+ RNAs signal to levels comparable to control HEK293T cells (Fig 4D). These results strongly support the conclusion that uridylated RNAs identified by the D391N CLIP-seq analysis are direct targets of DIS3L2 and that DIS3L2 is involved in their decay.

Next, we aimed to investigate what leads to the production of the uridylated 5′ mRNA fragments. We noticed that the following features of U+ 5′ mRfs are shared with the so-called transcription start site-associated short RNAs (TSSas): (i) close proximity to mRNA transcription start sites (TSS), (ii) frequent spanning to introns, and (iii) binding of DIS3L2 to U+ 5′ mRf did not correlate with changes in respective mRNA expression upon DIS3L2 inactivation (Fig EV4C). TSSas arise from both up and downstream of human bidirectional RNAPII promoters, as a result of RNAPII stalling and premature transcription termination (Seila *et al*, 2008; Taft *et al*, 2009; Valen *et al*, 2011; Ntini *et al*, 2013). D391N CLIPed U+ reads revealed peaks at positions +50 in sense and 200 in antisense orientation from mRNA TSS. These coincide precisely with the centers of peaks of stalling RNAPII (Fig 4E) and have identical location with previously reported 3′ ends of TSSa (Seila *et al*, 2008). We next performed subcellular fractionation of D391-oex cell line (Fig 4F) to test whether TSSa-like U+ 5′ mRfs are targeted by DIS3L2 in the cytoplasm. The RT–PCR analysis of RNAs isolated from nuclear and cytoplasmic fractions revealed strong accumulation of OAT and RPL12 5′ mRfs in the cytoplasm. The RT–PCR

amplification of the body of 5S rRNA was used as a positive control for RNA isolation and RT reaction in both fractions. The subsequent sequencing of several clones revealed that OAT 5′ mRfs amplified from the cytoplasmic, but not nuclear factions were uridylated (Fig 4F). In summary, our data demonstrated that at least a subset of TSSa RNAs are targets of the TDS pathway and that DIS3L2 is directly involved in their turnover in the cytoplasm.

## Discussion

Eukaryotic cells have evolved a number of mechanisms for RNA quality control, which come into play when faulty transcription or processing leads to aberrant RNAs. Here, we reveal for the first time an extensive catalog of oligouridylated mammalian RNAs. Moreover, we show that TUTase-DIS3L2 Surveillance (TDS) is a general pathway of cytoplasmic quality control for structured, mammalian ncRNAs. Uridylation and DIS3L2 were previously implicated in the rapid mRNA turnover in response to various stresses, such as viral infections or apoptosis (Abernathy *et al*, 2015; Eckwahl *et al*, 2015; Thomas *et al*, 2015; Haas *et al*, 2016). Here, however, we demonstrate that TDS is a constitutive machine that monitors an extensive repertoire of short as well as long ncRNAs in normal, unstressed conditions. In the yeast nucleus, an analogous TRAMP-exosome pathway controls degradation of a wide range of aberrant transcripts. We thus propose that eukaryotes possess subcellular compartment-specific mechanisms to deal with non-functional RNAs, both involving the tagging of target RNAs with homo-oligonucleotide runs. Nuclear surveillance involves oligo(A) tagging, whereas cytoplasmic utilizes oligouridylation.

The DIS3L2-bound uridylome revealed that TDS deals with many types of RNAs which are incompletely processed. Importantly, we captured U+ transcripts from a number of pseudogenes. We carefully inspected the mapping of U+ reads to these loci to ensure their correct annotation as pseudogenes. For 5S rRNA and other rRNAs, which occur in multiple genomic copies, assigning the reads to the appropriate loci is problematic. Nevertheless, after detailed manual

---

**Figure 4.   DIS3L2 targets short uridylated transcripts originating from mRNA transcription start sites.**

A    DIS3L2-bound uridylated mRNA reads map to the 5′ and 3′ UTRs.

B    Most 3′ UTR U+ reads match the stem-loop structure of histone mRNAs. Heat map representation of the uridylation position.

C    The uridylated aberrant RNAs are stabilized in cells with non-functional DIS3L2. Small RNAs were modified with 3′-end linker, and linker-specific primer was used for cDNA synthesis. Uridylated RNAs were PCR amplified in a semiquantitative way with the same amount of input RNA used for the RT reaction. We used a combination of the gene- and linker-specific primers, −RT ctrl is RT–PCR control, where no reverse transcriptase was added; PCR control is a reaction with no cDNA added. The graph on the right summarizes the percentage of uridylated and non-uridylated 5′ fragments of OAT mRNA in HEK293T-Rex cells (Ctrl) and cells overexpressing mutant DIS3L2 (D391N). Bands marked with * are unspecific primer–dimer PCR products.

D    DIS3L2-bound uridylated RNAs are degraded by DIS3L2 *in vivo*. RT–PCR amplification of 5′ mRfs of OAT and RPL12 genes and 3′-extended forms of U12 snRNA (RNU12 ext) from HEK293T cell lines with modified expression of DIS3L2 (as marked on top) was performed as described in (C). Regions of mature 5S rRNA and 7SL RNA were used as loading controls. DIS3L2 KO is a DIS3L2 knockout cell line. DIS3L2 rescue is DIS3L2 KO cells with stably integrated mutant (D391N) and wild-type (WT) DIS3L2 expressing constructs, respectively. Bands marked with * are unspecific primer–dimer PCR products.

E    Metagene analysis of the position of DIS3L2 D391N clipped U+ reads peak maximum around mRNA transcription start sites (TSS) and of the ChIP-seq data for RNAP II.

F    U+ 5′ mRfs and aberrant forms of U12 snRNA are highly enriched in the cytoplasm. RT–PCR detection of 5′ mRfs of the OAT and RPL12 and aberrant form of U12 snRNA was performed as described in (C) with RNA isolated from nuclear and cytoplasmic fractions, respectively. The RT–PCR amplification of the body of 5S rRNA was used as a positive control for RNA isolation and RT reaction in both fractions. The identity and uridylation status of cytoplasmic OAT U+ 5′ mRfs was confirmed by sequencing (shown on the bottom). In red are untemplated nucleotides, in green region corresponding to the first intron of OAT pre-mRNA. The efficiency of the subcellular fractionation was monitored by Western blot (right panel, WB). DDX5 is nucleoplasmic, ACO2 is mitochondrial, and DIS3L2 and TUBα are cytoplasmic markers.

Source data are available online for this figure.

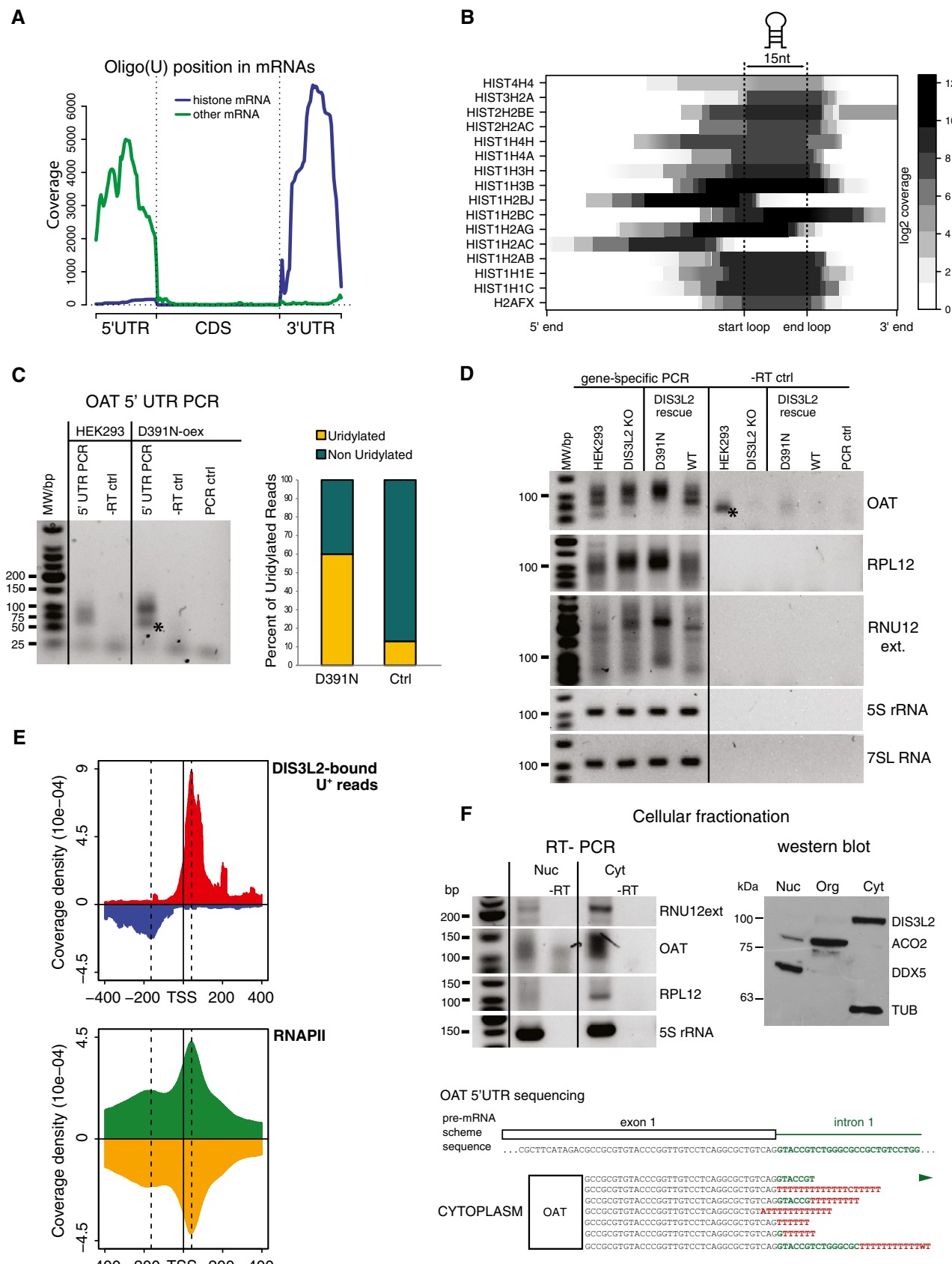

**Figure 4.**

inspection of 5S rRNA alignments, we concluded that TDS indeed targets transcripts from pseudogenes. The unifying feature of the DIS3L2-bound RNAs is that they are uridylated at positions that are close to stable secondary structures. We could even capture these as chimeric reads generated through intramolecular ligations during the CLIP protocol. The ligations occurred next to 5′ G nucleotides, reflecting the specificity of the RNase T1 treatment during CLIP. Although the captured intramolecular contacts mostly corresponded to structural elements present in mature RNAs, we also obtained hybrids reflecting tertiary contacts in tRNAs. This implies that aberrant RNAs mostly acquire known conformations and that TUTase/s are capable of adding UMPs to highly structured molecules. In some cases, DIS3L2 may target aberrant species independently of post-transcriptional uridylation. In our data, we are not able to distinguish untemplated extensions from encoded 3′-terminal Us of RNAPIII transcripts. However, the length of oligo(U) stretches on these transcripts rarely extended beyond the genome-encoded U tract. Thus, it is possible that terminators may serve as a signal for DIS3L2 targeting. During the revision process of this manuscript, a study was published also showing that uridylation and DIS3L2 mediate quality control of non-coding RNAs, such as snRNAs or Y RNAs (Labno *et al*, 2016).

Recent reports implicated monouridylation and stress-induced oligouridylation in mammalian mRNA turnover (Abernathy *et al*, 2015; Kim *et al*, 2015; Thomas *et al*, 2015). Our study however revealed yet another role of TDS in decay of transcripts from protein-coding genes. We uncovered an interesting set of uridylated forms of sense and antisense promoter proximal transcripts known as TSSas or tiny RNAs (Seila *et al*, 2008; Taft *et al*, 2009). Surprisingly, TSSas are not degraded by exosomes (Valen *et al*, 2011; Ntini *et al*, 2013). Therefore, TUT-DIS3L2 appears to be the main degradation mechanism for these short transcripts. Our D391N DIS3L2 CLIP also broadened the repertoire of uridylated pre-miRNAs. Most of the non-let-7 pre-miRNAs seem to lack any LIN28A-binding motif, which suggests that other cofactors facilitate their oligouridylation. The two most represented pre-miRNAs other than let-7 were miR-484 and miR-320. Interestingly, these pre-miRNAs are processed in a Drosha-independent manner from TSS-proximal prematurely terminated transcripts of protein-coding genes (Xie *et al*, 2013; Kim *et al*, 2016). Although we cannot exclude the possibility that miR-484 and miR-320 mapping U+ reads are in fact TSSas, these findings suggest a novel role for TDS in the regulation of alternative miRNA processing pathways.

# Materials and Methods

### Cross-linking and Immunoprecipitation (CLIP)

For the CLIP analysis, we used the previously described stable cell line of human embryonic kidney cells (HEK293T-Rex, Invitrogen) with inducible expression of human catalytically inactive form of DIS3L2 D391N that possess C-terminal fusion 3xFlag tag (Ustianenko *et al*, 2013). The CLIP protocol was a modification of the protocol described in Ule *et al* (2005). The cells were grown in DMEM and DIS3L2-Flag expression was induced 12 h before harvesting. Cells were washed with 1× PBS and exposed to 400 mJ of 365 nm UV. Cells were collected, frozen in liquid nitrogen, and

stored at −80°C. The subsequent steps of the CLIP protocol were performed with minor changes as described in Martin *et al* (2012). Briefly, cells were lysed in lysis buffer (LB, containing 50 mM Tris pH 7.5, 0.5% Triton X-100, 150 mM NaCl, supplemented with 1 mM DTT, protease inhibitor cocktail (Roche), and RNase inhibitor RNAsin (Promega)) and the insoluble fraction was sediment by centrifugation. FLAG-tagged DIS3L2 was immunoprecipitated using anti-FLAG M2 monoclonal antibody (Sigma) coupled to Protein G Dynabeads (Invitrogen). Bound protein–RNA complexes were extensively washed with LB containing 800 mM NaCl. The extracts bound to the beads were then split in two halves. One-half was treated with 1 unit/ml and the other aliquot with 5 units/ml of RNase T1 (Ambion, AM2283) for 10 min at 22°C, both parallels were cooled on ice and subsequently pooled back together. The extracts bound to the beads were further treated with 2 units of alkaline phosphatase (Fast-AP, Fermentas). The cross-linked RNAs were radiolabeled with polynucleotide kinase (T4 PNK, NEB) and gamma-$^{32}$P ATP. The 5′ adaptor (5′-rGrUrUrCrArGrArGrUrUrCrU rArCrArGrUrCrCrGrArCrGrArUrC-3′) was ligated to the bound RNA with T4 RNA ligase (Fermentas) in buffer containing 25% PEG 8000 at 16°C overnight. Protein–RNA complexes were resolved on a 4–12% gradient SDS–PAGE (NuPAGE, Invitrogen), and the region corresponding to the region above migration position of DIS3L2-Flag was cut out from the gel and eluted with proteinase K-containing elution buffer (50 mM Tris pH 7.5, 50 mM NaCl, 10 mM EDTA, 2 M urea, 2 mg/ml proteinase K) at 50°C for 2 h. RNAs were then ligated to the 3′ adaptor (5′-rAppAGATCGGAAGAGCACACGTCT -NH2-3′). RNA was size fractionated on 8% polyacrylamide/8 M urea gel. RNA fragments of the length of 70–110 nt were excised and extracted from the gel. Reverse transcription was done with a 3′ primer (5′-AGACGTGTGCTCTTCCGATCT-3′) and Superscript III reverse transcriptase (Invitrogen). The resulting cDNA was used as a template for PCR amplification (5′ Primer: 5′-AATGATACGGCG ACCACCGAGATCTACACGTTCAGAGTTCTACAGTCCG-s-A-3′). The PCR products were sequenced on the Illumina 2000 sequencer.

### Bioinformatics analyses of the CLIPseq results

Mapping of DIS3L2 mutant CLIP replicates to the reference genome GRCh37/hg19 was performed with the Segemehl software (Hoffmann *et al*, 2009). After initial mapping, reads that could not be mapped were trimmed of 3′-terminal stretches of more than three Ts stretches and mapping was repeated. All reads that mapped uniquely to the genome at one of these two steps were compared with the reference genome to identify untemplated poly(T) tails. For more details and statistical analyses, see Appendix Supplementary Methods.

### Gene annotation

Based on the human genome annotation, reads that mapped to annotated regions of the genome were assigned to the following annotation categories (listed here in order of their priority in our annotation procedure): tRNA → miRNA → snRNA → 5S rRNA → miscRNA → snoRNA → mRNA → lincRNA. The tRNA and miRNA genes were annotated with miRBase (Kozomara & Griffiths-Jones, 2014) database for miRNA and GtRNAdb (Chan & Lowe, 2009) for tRNA. The basic set of GENCODE tracks (Harrow *et al*, 2012) were

used for the remainder of the gene annotations. To allow annotation of potential pre-tRNA transcripts, tRNA genes were extended by 60 nt upstream and downstream of mature tRNAs.

## Immunoprecipitaion of FLAG-tagged DIS3L2 from human cells

Cells were resuspended in 4 ml of ice-cold lysis buffer (50 mM Tris pH 8.0, 150 mM NaCl, 0.5% Triton X-100 and Complete Protease Inhibitor Cocktail (Roche)) and incubated rocking at 4°C for 15 min. Lysate was cleared by centrifugation (10,000 g, 30 min, 4°C). For purification of FLAG-DIS3L2, 100 μl of anti-FLAG M2 beads (Sigma-Aldrich) washed with lysis buffer was incubated with cell extracts for 1 h in a cold room rotating. Beads were extensively washed with 10 volumes of wash buffer (50 mM Tris pH 8.0, 300 mM NaCl, 0.1% Triton X-100). Depending on the downstream application, proteins were eluted with one volume of 3×FLAG peptide (Sigma-Aldrich) resuspended in lysis buffer or by boiling with SDS loading buffer for 5 min.

## RNA immunoprecipitation

Cells were grown to 80% confluence, washed with ice-cold PBS, and UV cross-linked (400 mJ, 254 nm). Cells were lysed in buffer containing 150 mM NaCl, 50 mM Tris pH 7.6, 0.5% Triton X-100, supplemented with protease inhibitors (EDTA-free Complete Protease Inhibitor Cocktail, Roche), 0.5 mM EDTA, 1 mM DTT, RNasin (Promega). Lysates were cleared by centrifugation, and supernatants were applied on FLAG M2 Magnetic beads (Sigma) and incubated for 60 min. The bound fractions were washed two times with lysis buffer containing 150 mM NaCl and two times with 300 mM NaCl. RNA was eluted by treating whole beads with 2 mg/ml proteinase K (NEB) for 120 min at 37°C and subsequently phenol/chloroform extracted and ethanol precipitated.

## Preparation of DIS3L2 knockout cell line by CRISPR/Cas9

DIS3L2 knockout (K.O.) HEK293T cell line was prepared by using CRISPR/Cas9 technology as described in Ran *et al* (2013). Briefly, CRISPR guide sequences targeting the *DIS3L2* gene (A: CGGAGGT TCATTCTGTAGTCAGG, B: GGACCCCCAGAGGTAGTAAAAGG, C: CCAATAATGAGCCATCCTGACTA, D: CCAATAATGAGCCATCCTG ACTA) were cloned in pSpCas9n vector. We applied the double nickase technology using several pairs of CRISPRs to reduce the amount of off-target effects (pair 1—AB guides, pair 2—BC guides, pair 3—CD guides). The disruption of the open reading frame in mutant *DIS3L2* mRNA was confirmed by direct measurement of DIS3L2 protein expression by Western blot.

## Preparation of cDNA libraries for RNAseq analysis

The control HEK293T-Rex FlipIn cells, and cells overexpressing WT or mutant DIS3L2, and DIS3 K.O., were maintained in Dulbecco's modified Eagle's medium (DMEM), supplemented with 10% fetal calf serum at 37°C in the presence of 5% $CO_2$. Overexpression of Dis3L2 WT and DIS3L2 MUT were induced with 100 ng/ml of doxycycline for 24 h. Total RNA was isolated with TriPure Isolation Reagent (Roche) according to manufacturer's instructions, followed by RNase-free TURBO DNase (Ambion) treatment. Ribosomal RNA

was depleted from 4 μg of total RNA treated with TURBO DNase using the RiboMinus™ Eukaryote System v2 (Ambion) following the manufacturer's procedure. Efficiency of rRNA removal and concentration of rRNA-depleted RNA was monitored by Bioanalyzer 2100 (Agilent Technologies) using RNA 6000 Pico Kit. Sequencing libraries for whole transcriptome analysis were prepared using ScriptSeq™ v2 RNA-Seq Library Preparation Kit (Epicentre Biotechnologies) following the manufacturer's procedure. After the 3′-terminal tagging, the di-tagged cDNA was purified using MinElute PCR Purification Kit (Qiagen). Three 6-base index sequences were used to prepare bar-coded libraries for sample multiplexing (Script-Seq™ Index PCR Primers (Set 1); Epicentre Biotechnologies). PCR was carried out through 15 cycles to generate the second strand of cDNA, incorporate barcodes, and amplify libraries. A concluding Bioanalyzer 2100 run with the High Sensitivity DNA Kit (Agilent Technologies) that allows the analysis of DNA libraries regarding size, purity, and concentration completed the workflow of library preparation. The three RNA-seq libraries were then multiplexed in a single lane and sequenced on an Illumina HiSeq2000 using 50-bp single-end sequencing chemistry.

## RNA-seq analysis

Raw reads were subjected to adapter trimming using the tool cutadapt (Martin, 2011). Processed reads were aligned to the human transcriptome (hg38/GRCh38; ENSEMBL; merged sequences from protein-coding and non-coding RNA transcripts) (Cunningham *et al*, 2015) using the Segemehl software (Hoffmann *et al*, 2009). The estimation of the transcript expression as well as the differential expression (DE) analysis was done using the BitSeq software package (Glaus *et al*, 2012).

## Northern blot analysis

Total RNA was resolved on 10% denaturing polyacrylamide gel and transferred to Hybond-N+ membrane (GE Healthcare) by electroblotting (Bio-Rad). The hybridization with radioactively labeled oligonucleotides was performed in ULTRAhyb-oligo hybridization buffer (Ambion) at 38°C. Prior to addition of the labeled probe, the membrane was prehybridized at 42°C for 2 h. The radioactive signal was monitored by phosphorimager FLA-9000 (FUJIFILM). Quantification of signals was done using Multi Gauge software v3.2 (FUJIFILM).

## RNA isolation and PCR amplification of 5′ mRNA fragments (5′ UTR PCR)

Small RNA was isolated from indicated cell lines grown to 80–90% confluence treated with doxycycline (100 ng/ml of medium) using TriPure Isolation Reagent (Roche) followed by selective RNA precipitation. Fraction of RNA larger than 500 nt was first precipitated with 0.5 volume of 75% EtOH at room temperature for 10 min, followed by centrifugation (12,000 g, 4°C, 8 min.). The remaining small RNA in the supernatant was then precipitated with 0.7 volumes of isopropanol at 4°C for 30 min. followed by centrifugation (12,000 g, 4°C, 30 min). Small RNA fraction was treated with Turbo DNase (Ambion) extracted by phenol/chloroform and precipitated by isopropanol. 3′ pre-adenylated DNA linker ligation

was performed using T4 RNA Ligase2, truncated K227Q (New England BioLabs) according to manufacturer's instructions. Briefly, 2 μg of small RNA was incubated with 2 μl of 10 μM L3AppDNA linker (5rApp/AGATCGGAAGAGCACACGTCT/3ddC, IDT) and 200 U of the ligase in 20 μl of total reaction volume at 16°C overnight. RNA was reverse-transcribed with SuperScript RTIII (Invitrogen) using RT-CLIP2 primer and PCR amplified with gene-specific primers (Appendix Table S2). To make the PCR semi-quantitative and comparable between different cell lines, same amounts of RNA were used for linker ligation step and master mixes were used whenever possible.

### PCR amplification of snRNAs

After RNA immunoprecipitation, the isolated RNA was ligated with L3AppDNA linker (as in 5′ UTR PCR) and reverse-transcribed for 50 min. at 60°C using TGIRT-III (InGex) and RT-CLIP2 primer following manufacturer's instructions. PCR was performed with RT-CLIP2 and indicated gene-specific primers (Appendix Table S2).

### Subcellular fractionation

Subcellular fractionation was performed with minor modifications as described in Holden & Horton (2009) and Liu & Fagotto (2011). HEK293T cells (or modified as indicated in particular experiment) were grown to 75% confluency on 10-cm dish and induced with 100 ng/ml of doxycycline for 24 h. Cells were washed with cold 1× PBS on the dish and permeabilized with digitonin (in solution containing 45 μg/ml digitonin, 10 mM DTT, and 10 mM MgCl$_2$ in 1× NEH buffer) and incubated gently rocking at 4°C for 10 min. The released cytoplasmic fraction was collected from the dish and cleared by centrifugation at 500 *g* for 3 min. 1.5 ml TRI-reagent LS (Sigma) was added to 500 μl of the supernatant for subsequent isolation of the small RNA by selective precipitation as described above. Residues of the cells on the dish were washed with ice-cold 1× PBS and collected by scrapping in 1 ml of Buffer 2 (150 mM NaCl, 50 mM HEPES pH 7.4, 1% NP-40). After 30-min incubation on ice, nuclei were pelleted by centrifugation at 7,000 *g* for 15 min. The supernatant, which contained other organelles, was transferred to a new tube. The pellet was washed with cold 1× PBS and kept as the nuclear fraction. An aliquot from each fraction was used for a Western blot analysis. The cytoplasmic, nuclear, and organellar fractions were then used for small RNA isolation by selective precipitation as described above.

### Antibodies used in this work

Anti-FLAG M2 mouse monoclonal Ab, Sigma-Aldrich, cat. no. F3165; anti-alpha-tubulin mouse monoclonal Ab, Sigma, cat. no. T6074; anti-ACO2 (D6D9) XP rabbit monoclonal Ab, Cell Signalling Technologies, cat. no. 6571; anti-DDX5 (D15E10) XP rabbit monoclonal Ab, Cell Signalling Technologies, cat. no. 9877; anti-rabbit IgG HRP polyclonal secondary Ab, Promega, cat. no. W4011; anti-mouse IgG HRP polyclonal secondary Ab, Promega, cat. no. W4021; anti-DIS3L2 rabbit polyclonal Ab, produced by Moravian Biotechnology, assay (WB) specific test in Ustianenko *et al* (2013).

### High-throughput sequencing accession code

Raw and processed data of CLIP and RNAseq experiments can be found in GEO database under accession code GSE81537.

**Expanded View** for this article is available online.

### Acknowledgements

The authors thank Leona Kledrowetzova for excellent technical assistance and all members of Vanacova laboratory for useful discussions. We thank Torben Jensen for DIS3 and Bernd-Joachim Benecke for TUT1 expression constructs, Georges Martin for the initial help with stable cell line preparation and CLIP protocol optimization, and Mary O'Connell for reading the manuscript. Computational resources were provided by the CESNET LM2015042 and the CERIT Scientific Cloud LM2015085, provided under the program "Projects of Projects of Large Research, Development, and Innovations Infrastructures". This work was supported by the Wellcome Trust (084316/B/07/Z to S.V.), by the Czech Science Foundation (16-21341S to S.V., and J.P. was supported by 305/11/1095), and by the Ministry of Education, Youth and Sports of the Czech Republic under the project CEITEC 2020 (LQ1601).

### Author contributions

SV conceived, designed and interpreted experiments, and supervised the project. DU performed, analyzed, and interpreted the CLIP-seq analyses, JP designed and performed the bioinformatics analyses, ZF performed CLIP-seq validation experiments of ncRNAs and TSSas and helped with the manuscripts preparation, LB performed initial bioinformatics analyses, DZ performed cell fractionation and subsequent RNA analyses, and AF prepared stable cell lines, RNAs, and cDNA libraries for high-throughput sequencing. MZ helped with the design and interpretation of bioinformatics analyses and manuscript preparation. SV, DU, and JP wrote the manuscript.

### Conflict of interest

The authors declare that they have no conflict of interest.

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
