## [Review Process File · The EMBO Journal]

Manuscript EMBO-2016-94857

TUT-DIS3L2 is a mammalian surveillance pathway for aberrant structured non-coding RNAs

Dmytro Ustianenko, Josef Pasulka, Zuzana Feketova, Lukas Bednarik, Dagmar Zigackova, Andrea Fortova, Mihaela Zavolan and Stepanka Vanacova

Corresponding author: Stepanka Vanacova, Masaryk University

Review timeline:

Submission date:	25 May 2016
Editorial Decision:	20 June 2016
Revision received:	08 August 2016
Editorial Decision:	16 August 2016
Revision received:	18 August 2016
Accepted:	19 August 2016

Editor: Anne Nielsen

Transaction Report:

1st Editorial Decision

20 June 2016

Thank you for submitting your manuscript for consideration by the EMBO Journal. It has now been seen by three referees whose comments are shown below.

As you will see from the reports, our three referees all express great interest in the findings reported in your manuscript and would support publication here after adequate revision.

Given the referees' positive recommendations, I would thus like to invite you to submit a revised version of the manuscript, addressing the comments of all three reviewers. I should add that it is EMBO Journal policy to allow only a single round of revision, and acceptance of your manuscript will therefore depend on the completeness of your responses in this revised version.

You will see that the referees generally agree in the points they raise and for the revised manuscript I would therefore ask you to focus your efforts on the following issues:

-> Please provide additional data to address the specificity and validity of the U-rich RNAs associated with catalytically inactive Dis3L2 (ie points 1 and 2 from ref #1 and point 1 from ref #3)

-> Please also comment/expand on the nuclear-cytoplasmic distribution of the targets (refs # 1 and #3)

-> Ref #3 points out that the surveillance model remains partly correlative at the current stage and encourages you to include data on the consequence of Dis3L2 and TUT4 depletion for the RNA species found to interact with the catalytically inactive Dis3L2.

Thank you for the opportunity to consider your work for publication. I look forward to your revision.

REFEREE COMMENTS

Referee #1:

Earlier studies by these and other authors established that a variety of eukaryotic RNAs are tailed by 3' uridylation by terminal U-transferase (TUT) enzymes, and that 3' oligouridylation RNAs are preferentially degraded by the processive 3'-5' exonuclease DIS3L2. Loss-of-function mutations in DIS3L2 give rise to the Perlman fetal overgrowth syndrome, and have also been reported in sporadic Wilms' tumor. Earlier studies have focused on pre-let-7 microRNAs as substrates of the TUT-DIS3L2 RNA turnover pathway, as well as showing that a wide variety of mRNAs are TUT substrates. Here, the authors describe the results of UV cross-linking cellular RNAs to a catalytically inactive DIS3L2, a strategy designed to identify further DIS3L2 substrates in human cells. DIS3L2 targets identified in this way include mRNAs and pre-microRNAs, but also a wide variety of non-coding RNAs, particularly highly structured RNAs of this category. Interestingly, U-tailed, unspliced transcription start site-associated short RNAs (TSSasRNAs) were also found to be associated with catalytically inactive DIS3L2, suggesting that these products of abortive RNA polymerase II transcription are also turned over by TUT-DIS3L2 surveillance (termed here TDS). Together, these data provide further insight into the roles of DIS3L2, and hence the underlying molecular basis of Perlman syndrome and some aspects of Wilms' tumor, in a cell culture model. The data are for the most part of a high quality, and the clear, concise manuscript will in my view be suitable for publication if the authors are able to address the following points:

1. The central question is whether the RNAs purified by cross-linking to DIS3L2 D391N represent authentic substrates of wild type DIS3L2. The observation that DIS3L2 D391N can be cross-linked to Pol III transcripts with templated U-tails suggests that it may simply act as an affinity-purification 'sponge' for all RNAs bearing 3'oligo-U stretches, regardless of whether or not they are authentic DIS3L2 substrates. Further, all U-tailed reads mapping to 3' UTRs originated from terminal stem-loops of histone mRNAs, which have previously been reported by others to be substrates for ERI1, rather than DIS3L2. Can the authors distinguish DIS3L2 substrates among the larger category of RNAs that have (templated or untemplated) U-tails?
2. Figure 1: What is the level of expression of DIS3L2 D391N relative to endogenous DIS3L2? Does its expression induce a cell cycle phenotype similar to that previously described as resulting from DIS3L2 knock-down (Astuti et al., 2012)? If so, are any of the RNAs identified by CLIP affected by the cell cycle perturbation?
3. Figure 3: It is striking that wild type DIS3L2 stably associated with tRNA fragments of 24-30 nt. The authors suggest that these may represent the end products of DIS3L2-mediated degradation, but if this is the case it is not clear why they would remain associated with the exonuclease. Can the authors suggest a possible explanation?
4. Figure 4: The authors suggest that the TUT-DIS3L2 pathway is responsible for degrading TSSasRNAs that have been exported to the cytoplasm, but it seems equally or perhaps more likely that this is a nuclear process, suggesting that a fraction of TUT and DIS3L2 proteins reside in the nucleus. Can the authors address this possibility directly or, if not, suggest a mechanism by which TSSasRNAs could be exported to the cytoplasm?

Minor points:

As far as I am aware, 'uridylole' (Abstract and p8) is a neologism; 'uridylole' (if anything) might be more appropriate.

p3: 'GDL-2' should read 'GLD-2'.

Referee #2:

The manuscript by Ustianenko et al. explores the substrates of the ribonuclease Dis3l2, using CLIP-Seq coupled with a catalytically inactive variant of Dis3l2 that retains RNA binding activity. Importantly the authors show that the expression of the mutant does not really alter the cellular transcriptome and the majority of the CLIPed reads are uridylated; thus validating the experimental approach. The authors described the full compendium of cellular Dis3l2 targets uncovering many species of structured and aberrant ncRNAs but also short PolII-derived RNA products. The analysis of the CLIP-data set is meticulous with also the investigation of hybrid reads to support their observations and interpretations. Given Dis3l2 is a human disease and cancer locus, the dataset is important and provide possible insight in to disease mechanism. The statements and conclusions are supported by the data provided. The manuscript is well written and the presentation of the data is very accessible. I therefore fully support the manuscript for publication in EMBO.

Minor points:

1. Figure 4d is not referenced in the text and appears to be a repeat of Figure 1G.
2. I would exercise some caution of the identity of the Tut responsible for the uridylation of the identified Dis3l2 substrates. The experiments provided are in vitro. A statement in the text suggesting these observations are highly suggestive but require further validation could be prudent.

Referee #3:

In this manuscript, Ustianenko, et al. describe the RNA targets of the TUT-DIS3L2 decay pathway. DIS3L2 is an exonuclease with preferences for 3' terminal Us, which are most often added by cytoplasmic uridyl transferases (TUTs). Here, the authors take advantage of a catalytically inactive form of DIS3L2 to capture RNAs that have been targeted for degradation and identify these transcripts using high-throughput sequencing. This study is well conceived with good bioinformatics analysis, and the results highlighted here will be of interest to the community. However, additional experiments need to be performed to fully support their characterization of this surveillance pathway.

Major points:

- 1) The authors comment that 70% of the untrimmed CLIP reads did not map to the human genome. What fraction of these contained terminal Us? How many reads contained untemplated non-U residues?
- 2) Are the CLIP preparations cytoplasmically enriched? What fraction of the DIS3L2 D391N is cytoplasmic? This is especially important for the statement on pg 7 that "at least a subset of TSSa RNAs is exported to the cytoplasm." Without additional evidence, this sentence should be softened.
- 3) Although there is good circumstantial evidence for DIS3L2 mediating RNA surveillance, direct evidence of this is lacking from the current manuscript. What happens to the stability of the uridylated fragments in the absence of DIS3L2?
- 4) The in vitro analysis suggests that TUT4 may be responsible for much of the tRNA uridylation. What do the (in vivo) CLIP and RIP profiles of DIS3L2 look like when TUT4 is knocked down? How about TUT7 and GLD-2?

Minor points:

- 1) "Uridylome" should be "uridylome."
- 2) In the introduction, the authors cite Malecki, 2013 and Lim 2014 for the statement that "monouridylation of shortened poly(A) tails was implicated in mRNA turnover in yeast and human cells." Monouridylation of polyadenylated RNAs in yeast was originally identified in the lab of Chris Norbury.

We are grateful to all referees for their careful reading of our manuscript and for the insightful comments and suggestions. Where it was experimentally feasible, we addressed the referees' concerns and we believe that our revised manuscript will be suitable for publication in the EMBO Journal.

Referee #1:

Earlier studies by these and other authors established that a variety of eukaryotic RNAs are tailed by 3' uridylation by terminal U-transferase (TUT) enzymes, and that 3' oligouridylated RNAs are preferentially degraded by the processive 3'-5' exonuclease DIS3L2. Loss-of-function mutations in DIS3L2 give rise to the Perlman fetal overgrowth syndrome, and have also been reported in sporadic Wilms' tumor. Earlier studies have focused on pre-let-7 microRNAs as substrates of the TUT-DIS3L2 RNA turnover pathway, as well as showing that a wide variety of mRNAs are TUT substrates. Here, the authors describe the results of UV cross-linking cellular RNAs to a catalytically inactive DIS3L2, a strategy designed to identify further DIS3L2 substrates in human cells. DIS3L2 targets identified in this way include mRNAs and pre-microRNAs, but also a wide variety of non-coding RNAs, particularly highly structured RNAs of this category. Interestingly, U-tailed, unspliced transcription start site-associated short RNAs (TSSasRNAs) were also found to be associated with catalytically inactive DIS3L2, suggesting that these products of abortive RNA polymerase II transcription are also turned over by TUT-DIS3L2 surveillance (termed here TDS). Together, these data provide further insight into the roles of DIS3L2, and hence the underlying molecular basis of Perlman syndrome and some aspects of Wilms' tumor, in a cell culture model. The data are for the most part of a high quality, and the clear, concise manuscript will in my view be suitable for publication if the authors are able to address the following points:

1. The central question is whether the RNAs purified by cross-linking to DIS3L2 D391N represent authentic substrates of wild type DIS3L2. The observation that DIS3L2 D391N can be cross-linked to Pol III transcripts with templated U-tails suggests that it may simply act as an affinity-purification 'sponge' for all RNAs bearing 3' oligo-U stretches, regardless of whether or not they are authentic DIS3L2 substrates. Further, all U-tailed reads mapping to 3' UTRs originated from terminal stem-loops of histone mRNAs, which have previously been reported by others to be substrates for ERI1, rather than DIS3L2. Can the authors distinguish DIS3L2 substrates among the larger category of RNAs that have (templated or untemplated) U-tails?

We are grateful for this comment. We had initially similar concerns, however, several evidences from our work as well as from recent reports supported the conclusion, that majority of the identified uridylated RNAs are bona fide targets of DIS3L2. Our original experiments indicated, that uridylated fragments are stabilized upon overexpression of catalytically inactive DIS3L2 (Figure 4C). We have performed additional experiments to compare the abundance of U+5'mRFs and an example of the 3' extended (aberrant) U12 snRNA (RNU12) in cells with modified expression of DIS3L2. For that, we used the DIS3L2 KO cells and cells, in which we reintroduced mutant (D391N) or wild type (WT) DIS3L2. In the newly added panel Figure 4D, we demonstrate, that the abundance of uridylated 5'mRFs (of OAT and RPL12 genes) as well as an example of the 3' extended uridylated U12 snRNA (RNU12-ext) is increased upon DIS3L2 knock-out (KO). The abundance was greatly enhanced upon overexpression of D391N DIS3L2 mutants, whereas reintroduction of the wild type DIS3L2 in the KO cell line reverted the quantities of uridylated species to levels comparable to the control 293T cells. We believe, that these results strongly support the conclusion, that the DIS3L2-bound uridylated fragments and aberrant ncRNAs are physiological targets of DIS3L2.

Apart from our results, recent report of Eckwahl et al., 2015 showed, that DIS3L2 KD by siRNAs lead to increased levels of uridylated aberrant ncRNAs packed into virion particles in infected cells.

The new text accompanying results presented now as Figure 4C, 4D reads as the following:

"We were able to validate the existence of U⁺5'mRf independently of DIS3L2 binding by RT-PCR and sequencing of low molecular weight RNA isolated from control HEK293T-Rex cells (Figure 4C, S4A, S4B). We analyzed 30 clones of the OAT 5'mRfs PCR products from both control and D391N overexpressing cells. Sequencing analysis revealed that more than 60% of fragments in D391N-oex cells contained untemplated oligo(U) tails (Figure 4C), whereas only one clone from the control HEK293T cells had oligo(U) extension longer than three Us (Figure S4A). This demonstrated that overexpression of inactive DIS3L2 leads to U⁺5'mRfs upregulation. To further investigate the role of DIS3L2 in the turnover of U⁺5'mRfs, we prepared DIS3L2 knock-out (KO) cell line by using the CRISPR-Cas9 approach (Hsu, Lander et al., 2014, Ran, Hsu et al., 2013). To test to what extent is the DIS3L2 enzymatic activity involved in the turnover of U⁺ RNAs, we stably reintroduced the wild type (WT) or mutant (D391N) DIS3L2 to the KO cell line. As loading controls, we used PCR amplification products of the mature regions of 5S rRNA and 7SL RNA, because correctly processed functional ncRNAs do not appear to be targets of TDS. The levels of mature 5S rRNA and 7SL RNA remained unchanged in all cell lines tested. Notably, DIS3L2 depletion by KO resulted in accumulation of OAT and RPL12 U⁺5'mRfs as well as upregulation of the 3' extended forms of U12 snRNA (RNU12ext) (Figure 4D). This upregulation was strongly augmented by the additional overexpression of mutant DIS3L2 in KO cells. Conversely, reintroduction of the WT DIS3L2 resulted in reduction of the aberrant U⁺ RNAs signal to levels comparable to control HEK293T cells (Figure 4D). These results strongly support the conclusion, that uridylated RNAs identified by the D391N CLIP-seq analysis are direct targets of DIS3L2 and that DIS3L2 is involved in their decay."

The role of DIS3L2 in uridylated histone mRNA turnover still remains to be investigated in detail. It is true, that Eri1 exonuclease was implied as the main enzyme targeting uridylated histone mRNAs. DIS3L2 and ERI1 may have redundant roles in histone mRNA degradation. Alternatively, TUT-DIS3L2 may target primarily aberrantly processed histone mRNAs. The later is supported by the observation, that DIS3L2-bound uridylated reads map mostly to regions within the terminal stem-loop rather than at the mature 3' termini. Better understanding of the mechanism of DIS3L2 in uridylated histone mRNA turnover however will requires thorough investigations that are beyond the scope of this manuscript.

2. Figure 1: What is the level of expression of DIS3L2 D391N relative to endogenous DIS3L2? Does its expression induce a cell cycle phenotype similar to that previously described as resulting from DIS3L2 knock-down (Astuti et al., 2012)? If so, are any of the RNAs identified by CLIP affected by the cell cycle perturbation?

We were also initially concerned about the effect of mutant DIS3L2 expression in the cell line used for the CLIP analysis. The level of DIS3L2 overexpression upon doxycycline induction for 24 hours is approximately 20x higher than the level of the endogenous protein. Please not, that the cell line used for the CLIP analysis contained both; the endogenous DIS3L2 protein as well as the FLAG-D391N mutant. The expression of FLAG-D391N was induced only 24 hrs prior to the CLIP experiment. Before performing the CLIP analysis, we tested the effect of D391N overexpression on cell growth. As illustrated in the attached graph below, we did not observe any significant effect of D391N overexpression on cell growth compare to the control HEK293T cells. Moreover, the RNAseq analysis of the transcriptome profiles of the D391N oex cells did not reveal any distinct expression changes of the cell cycle factors (Supplementary Table 1). Therefore, we believe, that the RNAs identified by CLIP reflect the aberrant RNA species derived from RNA metabolism under "standard" conditions rather than due to cell cycle perturbation. We propose, that the mutant DIS3L2 overexpression mostly leads to the stabilization of these uridylated aberrant species, which allows their detection by CLIP.

Figure R1. The growth kinetics of the control HEK293T cell line and HEK293T cell line overexpressing FLAG-D391N DIS3L2 mutant. The results represent at least four independent tests.

3. *Figure 3: It is striking that wild type DIS3L2 stably associated with tRNA fragments of 24-30 nt. The authors suggest that these may represent the end products of DIS3L2-mediated degradation, but if this is the case it is not clear why they would remain associated with the exonuclease. Can the authors suggest a possible explanation?*

At this point, we do not have any explanation for the WT DIS3L2 association with the tRNA fragments. However, the results from the northern blot analyses correspond to our initial CLIP-seq analysis performed with the WT DIS3L2 (data not included as part of this manuscript). In that experiment we uncovered mostly fragments of tRNAs. One of our hypothesis was that DIS3L2 is involved in the production of the so called tRNA fragments (tRFs). However, to date, we were not able to satisfactorily validate this hypothesis.

4. *Figure 4: The authors suggest that the TUT-DIS3L2 pathway is responsible for degrading TSSasRNAs that have been exported to the cytoplasm, but it seems equally or perhaps more likely that this is a nuclear process, suggesting that a fraction of TUT and DIS3L2 proteins reside in the nucleus. Can the authors address this possibility directly or, if not, suggest a mechanism by which TSSasRNAs could be exported to the cytoplasm?*

We thank the referee for this interesting point. We performed subcellular fractionation followed by RNA isolation. We observed TSSas originating from the OAT gene strongly enhanced in the cytoplasmic fraction. Moreover, subcloning and sequencing revealed that only cytoplasmic fragments contain oligo(U) extensions. The new results are presented as Figure 4F and the newly added text in the last paragraph of the results reads as the following:

" We next performed subcellular fractionation of D391-oex cell line (Figure 4F) to test, whether TSSa-like U⁺5'mRfs are targeted by DIS3L2 in the cytoplasm. The RT-PCR analysis of RNAs isolated from nuclear and cytoplasmic fractions revealed strong accumulation of OAT 5'mRfs in the cytoplasm. The RT-PCR amplification of the body of 5S rRNA was used as a positive control for RNA isolation and RT reaction in both fractions. The subsequent sequencing of several clones revealed that OAT 5'mRfs amplified from the cytoplasmic, but not the nuclear fractions were uridylylated (Figure 4F). In summary, our data demonstrated that at least a subset of TSSa RNAs are targets of the TDS pathway and that DIS3L2 is directly involved in their turnover in the cytoplasm."

We also attempted to uncover the export pathway for TSSAs, however this appears to be complicated and is experimentally beyond the scope of this manuscript. Eckwahl et al., 2015 suggested, that aberrant ncRNAs that are packed into virions (such as tRNAs and snRNAs) are exported by exportin 5. However, currently it is premature to speculate if the same factors mediate also export of TSSAs.

Minor points:

As far as I am aware, 'uridylole' (Abstract and p8) is a neologism; 'uridylole' (if anything) might be more appropriate.

p3: 'GDL-2' should read 'GLD-2'.

We thank the referee for pointing to these mistakes. We corrected the text accordingly.

Referee #2:

The manuscript by Ustianenko et al. explores the substrates of the ribonuclease Dis3l2, using CLIP-Seq coupled with a catalytically inactive variant of Dis3l2 that retains RNA binding activity. Importantly the authors show that the expression of the mutant does not really alter the cellular transcriptome and the majority of the CLIPed reads are uridylylated; thus validating the experimental approach. The authors described the full compendium of cellular Dis3l2 targets uncovering many species of structured and aberrant ncRNAs but also short PolII-derived RNA products. The analysis of the CLIP-data set is meticulous with also the investigation of hybrid reads to support their observations and interpretations. Given Dis3l2 is a human disease and cancer locus, the dataset is important and provide possible insight in to disease mechanism. The statements and conclusions are supported by the data provided. The manuscript is well written and the presentation of the data is very accessible.

I therefore fully support the manuscript for publication in EMBO.

Minor points:

1. Figure 4d is not referenced in the text and appears to be a repeat of Figure 1G.

We want to thank the Referee No 2 for his positive feedback on the manuscript. The scatter plots in Figures 1G and 4D (now moved to Figure S4C) represent results of two different analyses. In Figure 1G we present the correlation between D391N-bound coding as well as ncRNAs and their differential expression between the D291N cell line used for the CLIP and the HEK293T cell line. In the second scatter plot (now as Figure S4C) we analyze only differential expression of mRNAs. The aim was to demonstrate, that overexpression of mutant DIS3L2 did not cause differential expression of protein coding genes with DIS3L2-bound 5' UTR fragments. The Figure S4C is now referenced in the text.

2. I would exercise some caution of the identity of the Tut responsible for the uridylation of the identified Dis3l2 substrates. The experiments provided are in vitro. A statement in the text suggesting these observations are highly suggestive but require further validation could be prudent.

We are aware of the fact that these results are only indicative. The identification of TUTase(s) involved in the TDS will require extensive studies, that are beyond the scope of this manuscript (see also our response to Referee No 3). We therefore changed the text accompanying the results from the in vitro assays as follows:

"To test, which of the known TUTases might be responsible for the uridylation of tRNAs, we compared the *in vitro* uridylation activities of purified TUT1, TUT4 and TUT7 using the tRNA fragment (tRF) as a substrate (Figure S3A). Whereas the activity of TUT1 was very weak, TUT7 catalyzed addition of long poly(U) tails. TUT4 modified the tRF with 10-20 UMPs, which most closely resembled the oligo(U) tails identified on RNAs in our CLIP data (Figure S3A). Moreover, the TUT4 activity enhanced the DIS3L2 degradation of tRFs *in vitro* (Figure S2B). These results

suggested, that TUT4 might be one of the enzymes acting in the TDS pathway. However, more extensive studies *in vivo* are needed to reveal the involvement of the individual TUTases in this surveillance pathway."

Referee #3:

In this manuscript, Ustianenko, et al. describe the RNA targets of the TUT-DIS3L2 decay pathway. DIS3L2 is an exonuclease with preferences for 3' terminal Us, which are most often added by cytoplasmic uridyl transferases (TUTs). Here, the authors take advantage of a catalytically inactive form of DIS3L2 to capture RNAs that have been targeted for degradation and identify these transcripts using high-throughput sequencing. This study is well conceived with good bioinformatics analysis, and the results highlighted here will be of interest to the community. However, additional experiments need to be performed to fully support their characterization of this surveillance pathway.

Major points:

1) The authors comment that 70% of the untrimmed CLIP reads did not map to the human genome. What fraction of these contained terminal Us? How many reads contained untemplated non-U residues?

Out of the 70% initially unmapped reads, 60% contained oligo(U) stretches longer than three nucleotides. Interestingly, around 15% of the unmapped reads possessed oligo(C) (see attached Figure R2). However, when we looked at the length distribution of the homonucleotide tails, the C-stretches showed a distinct peak at four nucleotides (see attached Figure R3). Because most of the C-tailed reads did not match to the genome after oligo(C) trimming, we excluded those reads from subsequent analysis. On the other hand, the tail length distribution of the oligo(U) stretches revealed a wide range of lengths, with the main enrichment around 8-9 nt (Figure R3).

Figure R3. The length distribution of homogeneous oligonucleotide

stretches at the 3'termini of the unmapped reads in the three CLIP experiments. The y axis shows number of reads in millions. The x axis shows the length of 3'terminal homogeneous oligonucleotide stretches.

2) Are the CLIP preparations cytoplasmically enriched? What fraction of the DIS3L2 D391N is cytoplasmic? This is especially important for the statement on pg 7 that "at least a subset of TSSa RNAs is exported to the cytoplasm." Without additional evidence, this sentence should be softened.

We thank the Referee for these important questions. We have performed subcellular fractionation, which showed cytoplasmic localization of DIS3L2 D391N (new Figure 4F). Moreover, the previous reports from us and others always demonstrated that DIS3L2 localizes to the cytoplasm (Ustianenko et al, RNA 2013, Lubas et al., EMBOJ 2013). We have have performed additional experiments, which demonstrated the cytoplasmic localization of 5' UTR mRNA fragments. Please see the response to the same question of Referee No. 1, comment No 4.

3) Although there is good circumstantial evidence for DIS3L2 mediating RNA surveillance, direct evidence of this is lacking from the current manuscript. What happens to the stability of the uridylylated fragments in the absence of DIS3L2?

We thank for this important question. Please, see our response to the same question to the Referee No1 above (Referee No1, point 1.).

4) The in vitro analysis suggests that TUT4 may be responsible for much of the tRNA uridylation. What do the (in vivo) CLIP and RIP profiles of DIS3L2 look like when TUT4 is knocked down? How about TUT7 and GLD-2?

This is a very interesting and important question and we are currently trying to address it. Previous works from Prof. Narry Kim, Prof. Bill Marzluft and others indicated, that mammalian TUTases have redundant functions. Therefore we need to perform an extensive set of experiments in order to be able to uncover to what extent are the individual enzymes involved in the TUT-DIS3L2 surveillance. Moreover, this will require new high-throughput analyses, which are beyond the scope of this work.

Minor points:

1) "Uridylom" should be "uridylome."

2) In the introduction, the authors cite Malecki, 2013 and Lim 2014 for the statement that "monouridylation of shortened poly(A) tails was implicated in mRNA turnover in yeast and human cells." Monouridylation of polyadenylated RNAs in yeast was originally identified in the lab of Chris Norbury.

We thank the referee for pointing to these mistakes. We have corrected the term "uridylo" to "uridylome" and added the missing reference to the work of Rissland and Norbury, 2009 to the text.

2nd Editorial Decision

16 August 2016

Thank you for submitting a revised version of your manuscript to The EMBO Journal. It has now been seen by one of the original referees whose comments are shown below. As you will see the referee finds that all major criticisms have been sufficiently addressed and recommends the manuscript for publication. However, before we can proceed to officially accept the manuscript there are a few editorial issues concerning text and figures that I need you to address.

I would therefore invite you to submit a final revision of your manuscript in which you include the following:

-> Please update figure call-outs in the text to fit the journal nomenclature (Appendix figure S1 etc)

-> Please include a Table of Contents at the first page of the appendix file.

-> The excel format of table S1 cannot be included in the appendix, please re-label and upload as Table EV1 (and refer to as such in the text).

-> We generally require that all information relevant to the main experiments in the manuscript should be included in Materials and Methods. I would therefore ask you to move the supplemental materials (at least the experimental sections, it's ok to leave the stats and basis for sequence mapping in the Appendix) into the main manuscript file.

-> We generally encourage the publication of source data, particularly for electrophoretic gels and blots, with the aim of making primary data more accessible and transparent to the reader. We would need 1 file per figure (which can be a composite of source data from several panels) in jpg, gif or PDF format, uploaded as "Source data files". The gels should be labelled with the appropriate figure/panel number, and should have molecular weight markers; further annotation would clearly be useful but is not essential. These files will be published online with the article as a supplementary "Source Data". Please let me know if you have any questions about this policy.

Thank you again for giving us the chance to consider your manuscript for The EMBO Journal and please contact me with any questions regarding the formatting. I look forward to your revision.

REFEREE COMMENTS

Referee #1:

The authors have satisfactorily addressed all the major points raised in my review of the earlier version of their manuscript. In particular, they have added supportive and persuasive data to strengthen their conclusions that: (1) the RNAs identified through association with catalytically inactive Dis3L2 are its authentic substrates; (2) expression of inactive Dis3L2 does not result in cell cycle perturbation and (3) U-tailed TSSasRNAs are present in the cytoplasm.

Corresponding Author Name: Stepanka Vanacova

Manuscript Number: EMBOJ-2016-94857